

# Oxytocin weakens self-other distinction in males during empathic responses to sadness: an event-related potentials study

Tong Yue, Ying Xu, Liming Xue and Xiting Huang

Faculty of Psychology, Southwest University, Chongqing, China

## ABSTRACT

By making use of event-related potential (ERP) technology, a randomized, double-blind, between-subject design study was performed in order to investigate whether OXT can weaken men's self-other distinction during empathic responses to sad expressions. In the two experimental tasks, 39 male subjects were asked to either evaluate the emotional state shown in a facial stimulus (other-task) or to evaluate their own emotional responses (self-task). The results revealed that OXT reduced the differences in P2 (150–200 ms) amplitudes between sad and neutral expressions in the self-task but enhanced P2 to sad expressions in the other-task, indicating OXT's role in integrating the self with others instead of separating them. In addition, OXT also reduced the LPC (400–600 ms) amplitudes between sad-neutral expressions in the self-task, implying that OXT's weakening effects on the self-other distinction could occur at both the early and late cognitive control stages of the empathic response.

## INTRODUCTION

Humans are social animals that rely strongly on social relations and interactions. The neuropeptide oxytocin (OXT), which is an ancient and conserved hormone, has been shown to influence many different aspects of social cognition and behavior in humans (*Baumgartner et al., 2008*; *Cardoso et al., 2012*; *Kumsta & Heinrichs, 2012*; *Quirin, Kuhl & Düsing, 2011*; *Sauer et al., 2013*). One particular concern, found in an increasing number of studies, is OXT's potential role in self-processing, especially the distinction between self and other. Some research have shown that OXT may sharpen the self-other percept by increasing the ability to discriminate between one's own face and unfamiliar faces, as observed during a morphing paradigm (*Colonnello et al., 2013*). On the other hand, other studies have indicated that OXT could blur the self-other distinction. For example, *Zhao et al. (2016)* found that OXT reduced self-bias in self and other trait judgments. This was reflected by reduced response times and accuracy of subsequent recall at the behavioral level, as well as reduced responses of the medial prefrontal cortex (mPFC) and its functional connectivity with the default mode network (DMN), which are involved in self-processing. All in all, the question of whether OXT blurs the distinction between self and other is still under debate.

Corresponding author
Tong Yue, yuetong87@swu.edu.cn

Empathy is an emotional state that results from the observation, imagination, and inference of emotions of other people, in conjunction with an awareness that a person's emotions originate within themselves (*De Vignemont & Singer, 2006*). The distinction between the self and others is regarded as an important cognitive component and emotional regulation strategy of empathy progression (*Decety & Jackson, 2004*). Whether from anecdotal observation of people's daily lives or from experimental research, men express significantly lower empathy than women (*O'Brien et al., 2013*; *Suh et al., 2012*), and some researchers believe that a higher tendency toward the self-other distinction may account for the lower empathy among men. For example, in the study of *Schulte-Rüther et al. (2008)*, participants were asked to perform either the self-task (evaluate one's affective responses to facial stimuli) or the other-task (focus on and evaluate the emotional states of facial expressions). The results showed that male subjects recruited the left temporoparietal junction (TPJ) more strongly compared with females in the self-task and that the TPJ is one component of a neural circuitry that plays a distinct role in self-other distinction (*Decety & Sommerville, 2003*; *Jackson et al., 2006*). Using similar self-tasks and other-tasks, The event-related potentials (ERPs) study of *Luo et al. (2015)* also reported that only males (but not females) exhibited an elevated P2 deflection when faced with sad expressions than neutral facial stimuli during the self-task; however, the discrepancy was absent during the other-task. According to the results, *Luo et al. (2015)* deduced that males might be inclined to distinguish the self from others during the progress of empathy, blocking their resonation with another individual's emotions. Therefore, the characteristics of male psychological processing with stronger self-other distinction in empathy progress provide a natural opportunity to test OXT's role in influencing self-processing: if OXT can effectively decrease the tendency toward self-other distinction in men, there would be an enhancing effect on empathy. On the other hand, OXT could weaken empathy in men or remain the same.

In fact, *Abu-Akel et al. (2015)* previously discussed the effects of OXT on empathy using the affective perspective-taking task, showing that OXT could increase empathy for pain in male subjects in the other-task instead of the self-task. According to the results, *Abu-Akel et al. (2015)* speculated that OXT may separate, rather than integrate, self and others in empathic responses. However, it may not be a compelling conclusion for several reasons. First, the indicator was not precise. *Abu-Akel et al. (2015)* adopted subjective reporting to measure the degree of empathic responses, but it was difficult to fully capture the differences in cognitive processing caused by variable manipulations solely under participants' introspection. Not only that, *Pfundmair et al. (2018)* found that OXT could blur the distinction between self and other only implicitly (e.g., reflexes and brain imaging) but not explicitly (e.g., ratings and self-report measures). Second, OXT's influence on increasing empathic responses when taking an other-perspective, but not a self-perspective, does not necessarily demonstrate its role in enhancing the ability to recognize differences between self and others. Using the visual perspective-taking task, *Yue et al. (2017)* reported that OXT can reduce self-centeredness tendencies and increase the perceptiveness of others. Thus, *Abu-Akel et al. (2015)* results can also be attributed to OXT's effects of promoting the transition from the self-to-other perspective and thus,

integrate the self with others instead of separating them. Therefore, it is necessary to verify this issue with more reliable and effective indicators.

Based on the above discussion, the present study assumes that OXT can effectively decrease the tendency toward self-other distinction in male empathic responses. In order to further determine whether the weakening effect occurs in the early stage of the empathic response's automatic processing or in the later stage of cognitive regulation, the present study adopts ERPs with higher sensitivity and time resolution. According to previous studies, the early P2 component and the late LPC component, which are associated with both self-related information (*Liu et al., 2013*) and processing of affective information (*Sheng & Han, 2012*; *Sheng et al., 2014*), can be effective indicators of the variation of self-other distinction in male participants' empathic responses. For example, it was found that P2 may reflect the gender difference in self-other distinction ability by the fact that sad faces induced larger P2 amplitudes in the self-task in men but not in women, since they have a greater tendency than women to automatically focus on and monitor their own emotion in response to distress cues in the early stage of empathic responses (*Luo et al., 2015*). Therefore, the first hypothesis of this study is, if OXT can decrease the male tendency toward self-other distinction in the early stage of the empathic response, the P2 discrepancy amplitude induced by the sad faces in the self-task should be absent, similar to that of female participants in the study of *Luo et al. (2015)*. In addition, LPC is an important indicator of self-other distinction ability for men, as they produced greater differential LPC amplitudes towards sad faces during the self-task compared with the other-task, while women did not (*Luo et al., 2015*). This is presumably because sadness can also communicate a request for help and elicit an approach and other-related prosocial motivations (*Seidel et al., 2010*), whereas men are more likely to allocate attentional resources toward self-related affective information caused by sad faces than women, which may underlie the gender difference of the LPC amplitude. Thus, if the effects of OXT last until the late processing stage of empathic responses, we can see signs from the LPC component, such as decreasing LPC amplitudes induced by sad faces in the self-task, which constitutes the second hypothesis of this study.

With reference to the previous articles (*Luo et al., 2015*; *Schulte-Rüther et al., 2008*), we employed two emotional perspective tasks using sad facial expressions to induce empathic responses in both the self-task and other-task. The present study employed a double-blind randomized controlled design, in which male participants in both the OXT and placebo (PLC) groups completed the emotional perspective tasks. By examining the influence of OXT on the P2 and LPC components of the empathic responses, this study explored whether it could decrease the male tendency toward self-other distinction in empathic responses.

## MATERIALS AND METHODS

### Participants and treatment

Forty male undergraduate students (mean age = 20.65 ± S.D. 1.85) were recruited via advertisements from Southwest University in China. Of these 40, one subject withdrew at the drug application stage because of rhinitis, leaving 39 subjects in total. Before the

experiment, each subject provided written informed consent. All the subjects were asked to maintain their regular sleep patterns before the formal experiment and to abstain from alcohol, caffeine, and smoking for at least 12 h. No participant had reported any neurological or psychiatric problems and none were taking any form of medication. The Ethics Committee of Southwest University approved the current study (IRB NO.H18007), and the procedures involved are in line with the Declaration of Helsinki.

We used a double-blind, controlled, and between-subject design in this study. First, each subject was given a single intranasal administration of 24 IU OXT or PLC. The OXT was administered in the form of a Syntocinon spray, produced by Sichuan Meike Pharmacy Co. Ltd, China; 3 puffs of 4 IU per nostril, with 30 s between each puff. The PLC, also produced by Meike Pharmacy Co. Ltd., comprised the same ingredients with the exception of the neuropeptide and was packaged in an identical bottle. *Striepens et al. (2013)* suggested that the formal experiment should start 45 minutes after administration of the OXT or PLC treatments because the peptide concentration would increase in the cerebrospinal fluid during this period. Ultimately, 20 men were treated with OXT, and the remaining 19 were treated with PLC (see Fig. 1). In the post-experiment interview, the subjects were unable to identify which treatment they had received (with any degree of accuracy better than by chance).

In order to control for the potential confounding effect of trait empathy between the OXT and PLC groups, before the start of the experiment, all subjects needed to complete an inter-personal reactivity index (IRI)-C (*Zhang et al., 2010*), which includes four sub-scales: fantasy (FS), perspective taking (PT), personal distress (PD), and empathic concern (EC).

## Experimental design

From the Chinese Affective Picture System (CAPS), we selected 30 sad and 30 neutral facial expressions, of which each category was divided evenly between males and females (*Bai et al., 2005*). According to the CAPS scores, $t$-test comparisons revealed significant differences between sad and neutral emotional faces on both valence (M ± SD: sad = 2.87 ± 0.68, neutral = 4.63 ± 0.79, $t(58) = -14.26$, $p < 0.001$) and arousal (sad = 5.68 ± 1.13, neutral = 3.70 ± 1.10, $t(58) = 7.62$, $p < 0.001$). All expressions were identical in size and resolution (15 cm × 10 cm, 100 pixels per inch). The pictures were presented at the center of a 17-inch CRT display, approximately 1 m in front of the subjects.

In the current study, we performed two emotional perspective tasks, which included both a self-task and an other-task. In the other-task, we told the subjects to press the F button if the emotion expressed by the face was sad and the J button if neutral. The order of the F (sad) and J (neutral) keys was counterbalanced between participants. The procedure of the self-task was similar to that of the other-task, except that we asked the subjects to evaluate their own emotional reaction when observing the presented faces in the experiment. The subjects were emphatically told that there were absolutely no right or wrong answers and instructed to make choices according to their own feelings.

Every task had two blocks, each containing 60 trials (30 emotional and 30 neutral faces). The order of the two tasks was randomized across subjects. After completing each block,

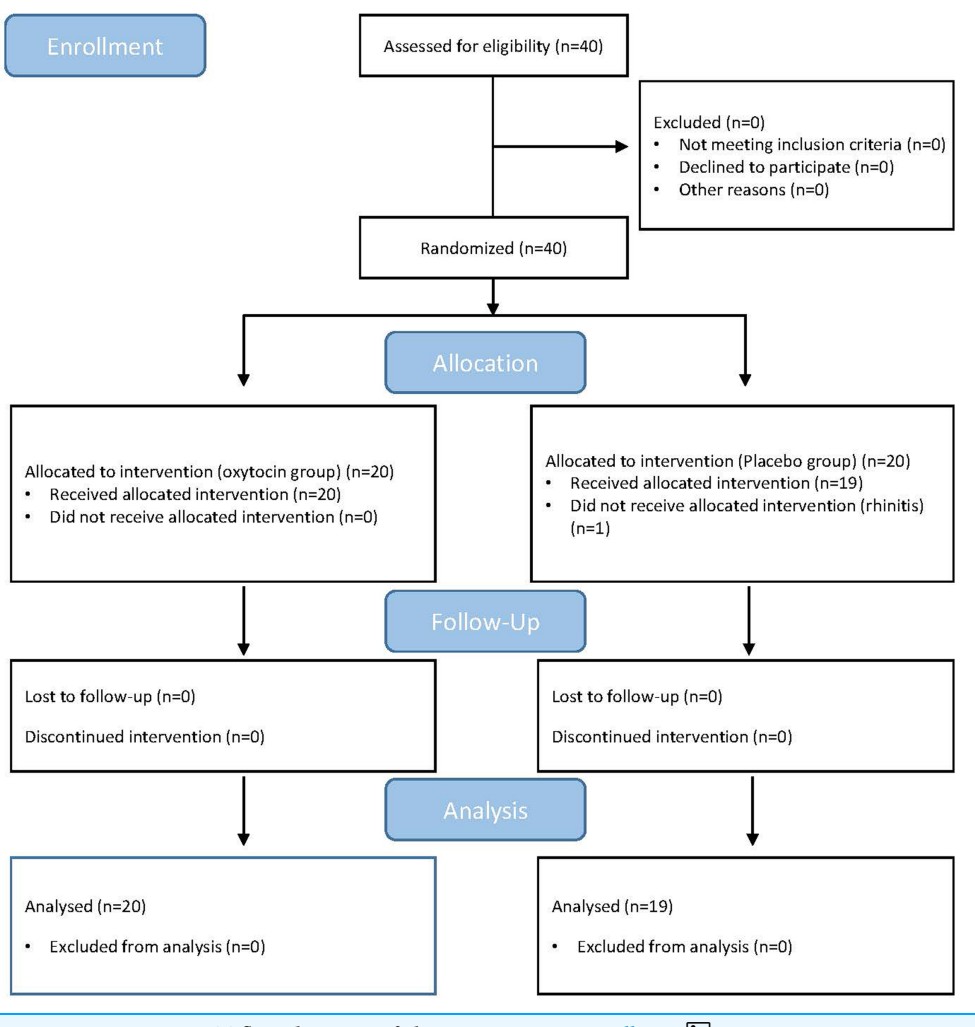

**Figure 1 CONSORT flow diagram of the participants.**

the participants were permitted to rest. Before the formal experiment, the subjects were instructed to familiarize themselves with the experimental procedure via practice trials. The trial in the formal experiment was conducted as follows: First, a red fixation "+" lasting 500 ms was presented on the black screen. Then, a blank screen was shown for 500–800 ms, followed by pictures of either sad or neutral faces, which were presented in random order. The subjects were asked to respond according to the requirements of the task as quickly and accurately as possible, and once the subject pressed a key, the facial expression on the screen was terminated. If no response was made, the visual stimulus disappeared automatically after 2,000 ms.

## EEG recording and data processing

The EEG data were recorded through 64 scalp sites with tin electrodes mounted in an elastic cap (Brain Products, Munich, Germany). The horizontal and vertical electrooculograms (EOGs) were recorded from electrodes placed above and at the outer canthus of the right eye to control for eye movement artifacts. The ground electrode was on
the medial frontal aspect and the references were on the left and right mastoids. The EEG and EOGs were amplified with a 0.05–100 Hz bandpass filter and a 500 Hz sampling rate. The electrode impedances were kept below 10 kΩ. Subjects were asked to stay still, avoid blinking, relax facial muscles, and look at the fixation cross on the screen.

We used the Vision Analyzer 2.1 (Brain Products, Gilching, Germany) to perform the off-line analyses. The data were segmented into 1,200 ms durations, including 200 ms of pre-stimulation baseline recording. The regression analysis was performed to remove the EOGs with a band pass filter of 0.01–30 Hz. The trials excluded from the averaging included trials with artifact contamination due to amplifier clipping and EOG artifacts (average EOG voltage exceeding ±80 mV), bursts of electromyography activity, and peak-to-peak deflection exceeding ±80 mV.

Based on visual observation, as well as on previous studies (*Sheng & Han, 2012*), we analyzed the mean amplitude of the 150–200 ms period after the onset of the outcome for the P2, and 400–600 ms for the LPC. The P2 was mainly distributed in the frontal lobe and central regions, while LPC was widely distributed on the scalp. Therefore, we included the mean amplitudes of electrodes located in the frontal (F3, F4, Fz, FC3, FC4, FCz) and central (C3, C4, Cz) regions for the P2 analysis, and the frontal (F3, F4, Fz, FC3, FC4, FCz), central (C3, C4, Cz), central parietal (CP3, CP4, CPz), and parietal (P3, P4, Pz) regions for the LPC analysis. Regarding the effect of laterality, we also took into account the electrodes on the left hemisphere (F3, FC3, C3, CP3, P3), central line (Fz, FCz, Cz, CPz, Pz), and the right hemisphere (F4, FC4, C4, CP4, P4) for the LPC analysis.

A mixed-design ANOVA was performed on the mean amplitudes of the P2 and LPC with brain regions, laterality (left hemisphere, central line and right hemisphere), facial expression (sad and neutral) and task (self and other) as the within-subjects factors and treatment type (OXT vs. PLC) as the between-subjects factor. The data were analyzed via SPSS 16.0 using the Greenhouse-Geisser correction when the sphericity hypothesis was violated, and Bonferroni correction was applied for multiple comparisons.

# RESULTS

## Behavioral results

The independent *t*-test of IRI-C scores showed that there was no significant difference between the OXT and PLC groups, regardless of the four sub-dimensions or the total dimensions (all $p > 0.51$).

Accuracy analysis revealed a significant main effect of task type ($F_{1, 37} = 47.72$, $p < 0.001$, $\eta_p^2 = 0.56$), with significantly greater accuracy during the other-task ($M \pm SD$: 89.43 ± 9.47%) than during the self-task (68.44 ± 20.93%). Another significant main effect of facial expression ($F_{1, 37} = 11.20$, $p < 0.01$, $\eta_p^2 = 0.23$) was discovered, showing that accuracy for the neutral faces (84.20 ± 22.00%) was significantly higher than for the sad faces (73.67 ± 18.19%). No other main effects or interactions were found to significantly impact the degree of accuracy. The ANOVA of response time revealed a significant main effect of task type ($F_{1, 37} = 15.90$, $p < 0.001$, $\eta_p^2 = 0.30$), with RTs being slower in the self-task (1,190.48 ± 246.97 ms), as compared to the other-task (990.66 ± 190.84 ms). The main effect of emotional types on response time was also significant ($F_{1, 37} = 9.92$, $p < 0.01$,

$\eta_p^2 = 0.21$), with longer RTs for sad expressions, as compared with neutral expressions (1,142.09 ± 237.19 vs 1,039.05 ± 232.47 ms). No other main effects or interaction were found in the response time analysis.

## ERP results

### P2

The ANOVA of the mean amplitudes of the P2 component suggested a main effect of affect ($F_{1, 37} = 4.11$, $p = 0.05$, $\eta_p^2 = 0.10$), with significantly larger amplitudes induced by sad faces (3.44 ± 0.49 µV) than by neutral faces (2.93 ± 0.52 µV). There was also a significant interaction effect involving treatment type, task, and facial expression ($F_{1, 37} = 5.99$, $p < 0.05$, $\eta_p^2 = 0.14$). Results of the simple effects test showed that sad expressions (3.05 ± 0.79 µV) elicited larger P2 amplitudes than did neutral expressions (1.79 ± 0.80 µV) in the placebo group's self-task ($F_{1, 37} = 5.84$, $p < 0.05$, $\eta_p^2 = 0.14$), but that there was no significant difference between the two facial expressions in the other-task. For the OXT group, sad faces (4.21 ± 0.70 µV) induced larger P2 amplitudes than did neutral faces (3.11 ± 0.77 µV) in the other-task ($F_{1, 37} = 4.32$, $p < 0.05$, $\eta_p^2 = 0.10$), while there was no difference in the self-task. In addition, significant interaction effects were found between brain area, laterality, treatment type, task, as well as facial expression (see Table 1). Further analysis showed that under the influence of OXT, sad faces elicited larger P2 amplitudes in the left central (C3) region in the other-task, while neutral faces induced larger P2 amplitudes in the left and right left central (C3, C4) regions in the self-task (see Fig. 2).

ANOVAs of the difference waves at the P2 component revealed a significant interaction effect between treatment type and task ($F_{1, 37} = 5.52$, $p < 0.05$, $\eta_p^2 = 0.13$), and the simple effects test showed no obvious difference between the two groups in the other-task, except that the P2 of the OXT group was significantly smaller than that of the PLC group in the self-task ($F_{1, 37} = 5.51$, $p < 0.05$, $\eta_p^2 = 0.13$) (see Fig. 3).

### LPC

The analyses of ERP amplitudes focusing on the LPC component showed that the main effects of facial expression, brain region, and laterality were all significant ($F_{1, 37} = 48.24$, $p < 0.001$, $\eta_p^2 = 0.56$; $F_{1, 37} = 63.18$, $p < 0.001$, $\eta_p^2 = 0.62$; $F_{1, 37} = 7.59$, $p = 0.001$, $\eta_p^2 = 0.17$) (see Table 1). Compared with neutral expressions, sad expressions induced larger LPCs (8.82 ± 0.73 µV vs 6.59 ± 0.73 µV). The amplitudes from the frontal-central to the parietal regions gradually decreased, and the amplitudes over the midline were larger than those of the left and right sides. Additionally, there was a significant three-way interaction for facial expression, task, and treatment type: for participants in the OXT group, larger LPC amplitudes were elicited by sad expressions, as compared with neutral faces in the other-task (8.30 ± 1.02 µV vs 6.01 ± 1.05 µV, $p < 0.01$), but with no significant difference in the self-task; for participants of the PLC group, sad expressions induced larger LPC amplitudes than neutral expressions in both the other-task (8.96 ± 1.07 µV vs 6.97 ± 1.10 µV) and the self-task (8.82 ± 1.45 µV vs 5.08 ± 1.16 µV; $p < 0.01$). Additionally, a significant four-way interaction for treatment type, task, brain region, and facial

**Table 1 Overall results of ANOVA on P2 (150-200 ms) and LPC (400–600 ms).**

| Source of variations | P2 | | LPC | |
|---|---|---|---|---|
| | $F$ | $\eta_p^2$ | $F$ | $\eta_p^2$ |
| Treatment type | — | — | — | — |
| Task | — | — | — | — |
| Facial expression | 4.11 | 0.10* | 48.24 | 0.56*** |
| Brain regions | — | — | 73.40 | 0.66*** |
| Laterality | 9.31 | 0.20*** | 7.59 | 0.17** |
| Treatment type × task | — | — | 10.16 | 0.21** |
| Treatment type × facial expression | — | — | — | — |
| Treatment type × brain regions | — | — | — | — |
| Treatment type × laterality | — | — | — | — |
| Task × facial expression | — | — | — | — |
| Task × brain regions | 6.77 | 0.13* | — | — |
| Task × laterality | — | — | — | — |
| Facial expression × brain regions | — | — | — | — |
| Facial expression × laterality | — | — | — | — |
| Brain regions × laterality | — | — | 10.37 | 0.21*** |
| Treatment type × task × facial expression | 5.99 | 0.14* | 6.49 | 0.15* |
| Treatment type × task × brain regions | — | — | — | — |
| Treatment type × task × laterality | 5.24 | 0.12** | — | — |
| Task × facial expression × brain regions | — | — | 7.36 | 0.16*** |
| Task × brain regions × laterality | 10.33 | 0.22*** | — | — |
| Facial expression × brain regions × laterality | 9.56 | 0.21*** | — | — |
| Treatment type × task × facial expression × brain regions | — | — | 2.83 | 0.07* |
| Treatment type × task × facial expression × laterality | — | — | — | — |
| Task × facial expression × brain regions × laterality | 4.70 | 0.11* | — | — |
| Treatment type × facial expression × brain regions × laterality | 5.90 | 0.14** | — | — |
| Treatment type × task × facial expression × brain regions × laterality | 9.40 | 0.20*** | — | — |

**Note:**
***$p < 0.01$, **$p < 0.01$, *$p < 0.05$, — non-significant.

expression was also found ($F_{1, 37} = 2.83$, $p < 0.05$, $\eta_p^2 = 0.07$), and further analysis indicated that under the impact of OXT, the LPC amplitudes induced by neutral expressions in the frontal regions were the largest.

The analysis of the difference waves at the LPC component revealed that the interaction effect between group and task was significant ($F_{1, 37} = 5.99$, $p < 0.05$, $\eta_p^2 = 0.14$). Further, the simple effects test showed that there was no significant difference between the two treatment types in the other-task, but smaller difference waves were found in the OXT group than in the PLC group in the self-task ($F_{1, 37} = 13.73$, $p = 0.001$, $\eta_p^2 = 0.28$) (see Fig. 4).

# DISCUSSION

In the present study, we explored whether OXT can weaken men's self-other distinction in the empathic responses during emotional perspective tasks. Although there was limited evidence at the behavioral level, at the more sensitive neural level, we found differences

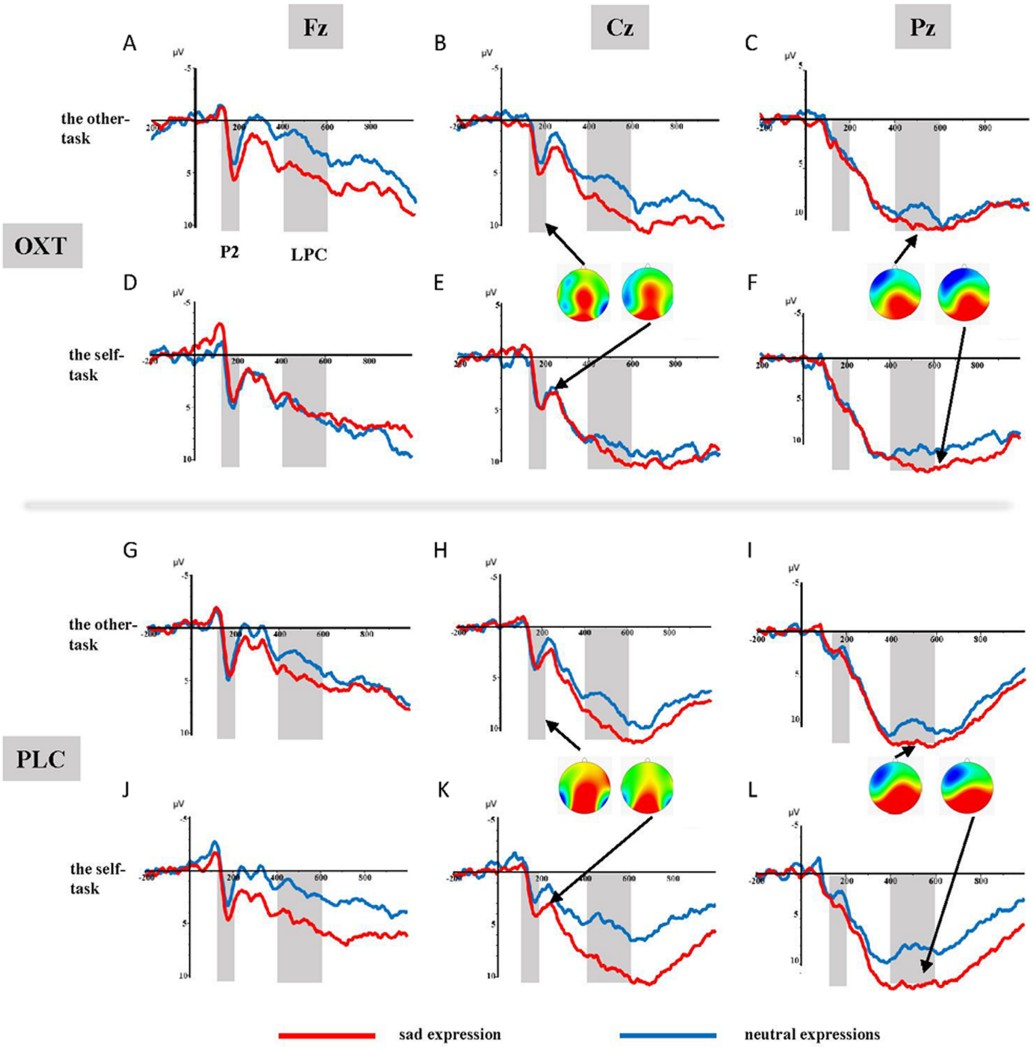

**Figure 2 (A–L) Grand-average ERP waveforms from Fz, Cz and Pz channels, elicited by emotional faces in the OXT and PLC groups for the other-task and the self-task.**

between the OXT and PLC groups reflected in the P2 and LPC components, which provided evidence that OXT blurs self-other distinction.

At the behavioral level, the difference in accuracy or reaction time between the OXT and PLC groups during the emotional perspective tasks was not significant, which is consistent with previous studies (*Luo et al., 2015*; *Jin et al., 2013*; *Schulte-Rüther et al., 2008*). Previous studies have also suggested that OXT could blur the self-other distinction only in an implicit response but not at an explicit level (*Pfundmair et al., 2018*). This may be because the explicit response is conscious, requires cognitive resources, and may even be superimposed by conscious thoughts or self-promotion, while the implicit response is not (*Bargh, 1994*). Thus, the explicit behavioral indicators may be less sensitive to the effects of OXT, and the findings of this study are mainly reflected at the level of implicit neural responses.

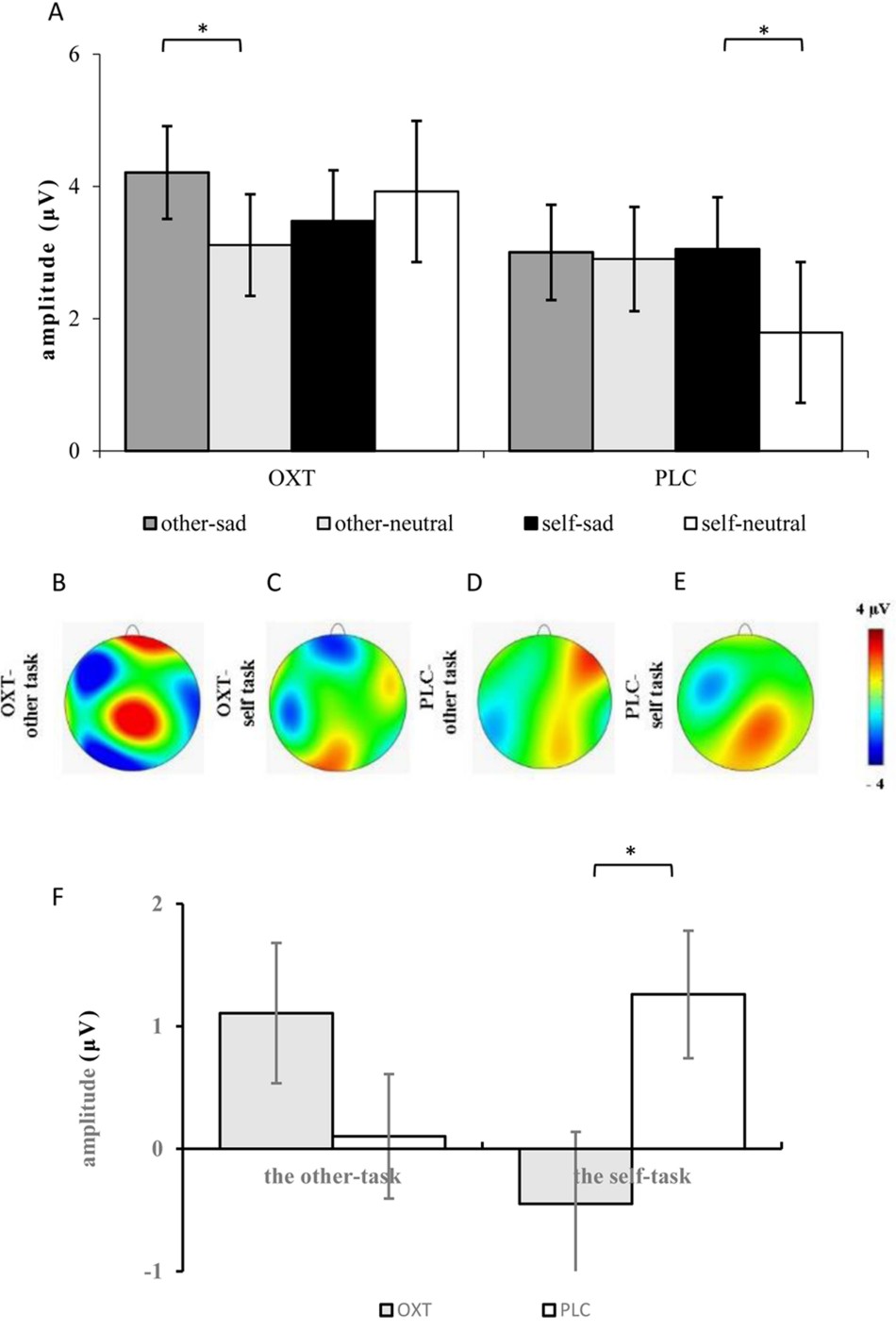

**Figure 3** **(A) Comparison of means and standard errors of P2 amplitudes. (B–E) Topographic maps elicited by the other- and the self-task. (F) The difference waves of P2 between the OXT and PLC groups.** Symbols indicate significance level (*$p < 0.05$).

The P2 results of the PLC group validated the conclusion of *Luo et al. (2015)*, who conducted tests on males and also reported that sad facial stimuli elicited larger amplitudes than did neutral facial stimuli during the self-task, but the discrepancy was absent during

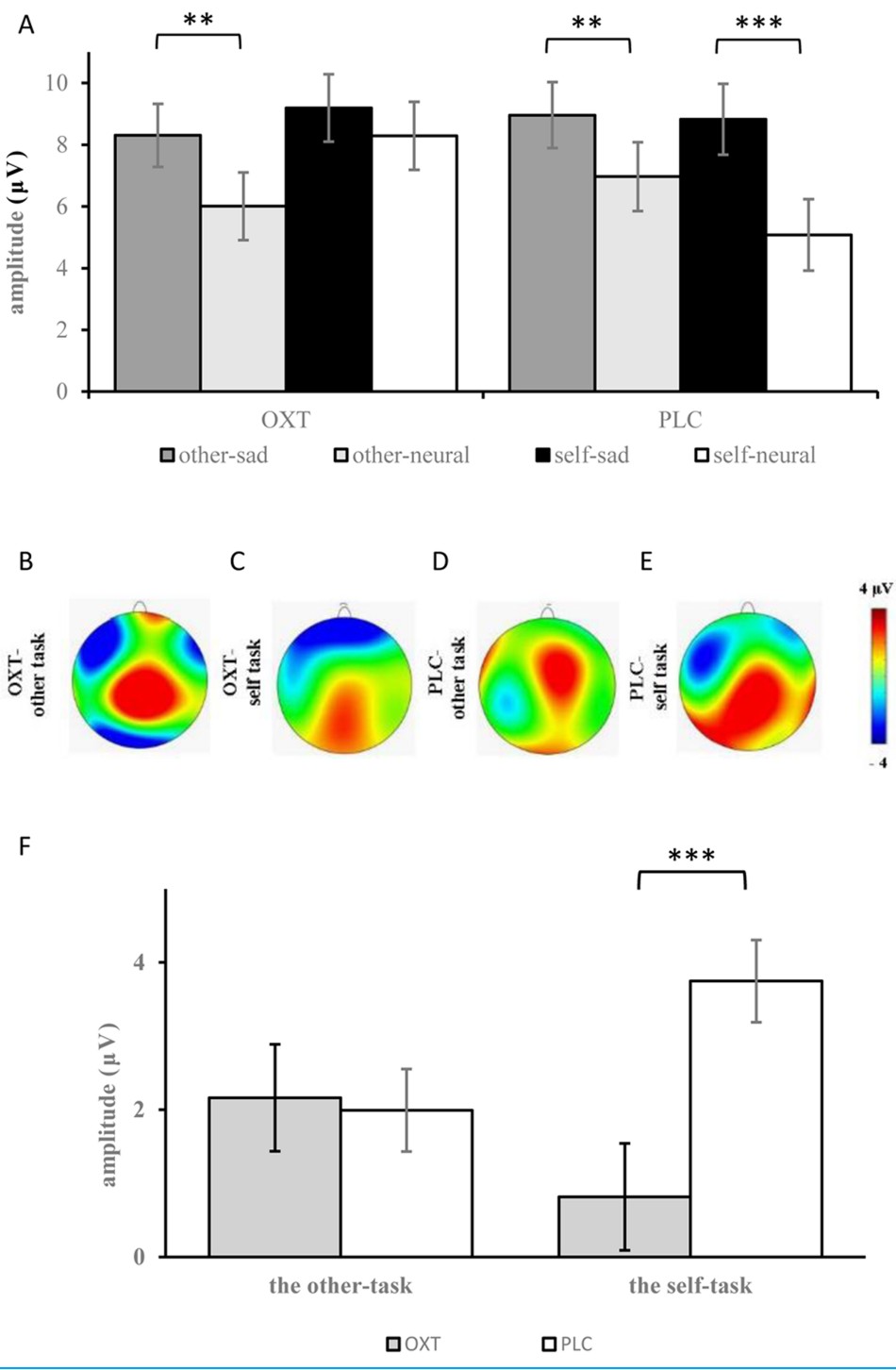

**Figure 4** **(A) Comparison of means and standard errors of LPC. (B–E) Topographic maps in the OXT and PLC groups. (F) The difference waves of the LPC between the OXT and PLC groups.** Symbols indicate significance level (*** $p < 0.001$, ** $p < 0.01$).     

the other-task. According to previous studies, the P2 component is sensitive to salient stimuli (*Carretié et al., 2001*; *Huang & Luo, 2006*; *Yuan et al., 2007*) and is therefore considered to be associated with early automatic affect sharing during empathic responses

(*Sheng & Han, 2012*; *Sheng et al., 2014*); on the other hand, it might also reflect the attentional effect in self-relevant information (*Liu et al., 2013*). Thus, the larger P2 amplitudes elicited by sad than neutral facial stimuli during the self-task in the PLC group may reflect the attentional enhancement caused by self-relevance processing in the early stage, which suggests that males are likely to spontaneously adopt the self-perspective, and attend more to their own feelings, despite the origin of these feelings from others. Therefore, we suggest that men's reactions to their own emotions in P2 may reflect their differences between themselves and others, which echoes the findings by *Schulte-Rüther et al. (2008)*.

These results support our first hypothesis that male participants in this study would show a trend consistent with female participants in the study of *Luo et al. (2015)* after OXT usage; that is to say, although sad facial stimuli induced larger P2 compared to neutral facial stimuli during the other-task, the difference between these two expressions disappeared in the self-task. Further analysis on the differential amplitudes showed that, compared with the PLC group, OXT significantly reduced the P2 differentiation between the two expressions in the self-task but tended to increase differences in P2 in the other-task. Although the weakening of the differential wave in the self-task could be attributed to decreasing brain reactivity to negative emotions and increasing neural activity involved in emotion regulation (*Ma et al., 2016*), it is difficult to explain the opposite trends in the other-task. Therefore, the more likely reason is OXT's established role in decreasing self-interest and increasing interest in others (*Yue et al., 2017*) that is reflected in the present study by a shift in male participants' focus from self to others, which weakens their emotion monitoring and regulation in response to facial expressions, leading to a differentiation between the facial stimuli at P2 arising in the other-task. In other words, OXT may play a role in attenuating the distinction between the self and other during the early automatic stage of the empathic response.

As for the LPC component, the results of the PLC group revealed that the mean amplitudes induced by the sad expressions were greater than those elicited by the neutral facial stimuli in both the self- and the other-task; however, for the OXT group, the difference between the sad and neutral expressions was only significant in the other- but not in the self-task. Further analysis on the difference waves of the LPC showed that in the self-task, they were significantly smaller in the OXT group than in the PLC group, but this trend did not show in the other-task. Similar to the P2 component, the LPC component is believed to reflect the attention distribution processing of emotion-related stimuli (*Li et al., 2012*; *Schupp et al., 2000*; *Van Strien, De Sonneville & Franken, 2010*) and may be related to the intrinsic motivational significance of self-related information (*Fields & Kuperberg, 2012*; *Li & Han, 2010*). Given that after inhaling OXT, the discrepancy between sad and neutral expressions remained only during the other-task but not the self-task, the OXT's effect cannot be attributed to downregulation of emotionally relevant stimuli (*Ma et al., 2016*) but rather to its effects of reducing self-orientation. Thus, the above results validate the present study's hypothesis that the weakening effect of OXT on men's self-other distinction still exists in the later stage of the empathic response.

 

Several limitations may still exist in our current study, which can be addressed in future studies. First, the current study only used sad and neutral faces as the empathy inducing stimuli. Thus, the subjects simply needed to choose one emotion. This design cannot explain whether the OXT effects observed in this study only apply to sad-specific emotions, general negative emotions, or whether they also apply to positive emotions. Future research should investigate whether other types of emotions could replicate the same OXT effects. Moreover, differences in sex are an important factor when examining the effects of OXT on human social cognition (*Fischer-Shofty, Levkovitz & Shamay-Tsoory, 2013*; *Preckel et al., 2014*; *Theodoridou, Penton-Voak & Rowe, 2013*). Thus, the potentially sexually dimorphic effects of OXT on the self-other distinction during empathy should be considered in future studies. Our conclusion that OXT weakens the self-other distinction during empathic responses was, to a large extent, based on indirect inferences. Future researchers could select more direct indicators (e.g., the activity of self-related brain areas), to further reveal the influence of OXT. Ultimately, the sample size in the current study is relatively small and we did not measure affective state/mood or the awareness of one's own emotions (self-awareness/alexithymia), which may affect the results of this study to some extent. Addressing these deficiencies would be an important avenue for future research.

## CONCLUSIONS

The results showed that OXT reduces the differences between neural responses to sad and neutral expressions in the self-task but enhances neural responses to sad expressions in the other-task during the early stage of the empathic response, which provides evidence that OXT appears to blur the distinction between self and other in the context of decreasing self-interest and increasing interest in others. In addition, OXT's effects of reducing the neural differences between sad and neutral expressions in the self-task persisted in the later cognitive control stage, which implies that the role of OXT in weakening the self-other distinction seems to occur throughout all stages of the empathic response.

### Funding

This work was supported by the PhD research startup foundation of Southwest University (SWU118092) and the Research Funds for the Central Universities (SWU1809212). The funders had no role in study design, data collection and analysis, decision to publish, or preparation of the manuscript.

### Grant Disclosures

The following grant information was disclosed by the authors:
Southwest University: SWU118092.
Research Funds for the Central Universities: SWU1809212.

## Competing Interests

The authors declare that they have no competing interests.

## Author Contributions

- Tong Yue conceived and designed the experiments, performed the experiments, analyzed the data, prepared figures and/or tables, authored or reviewed drafts of the paper, and approved the final draft.
- Ying Xu performed the experiments, analyzed the data, authored or reviewed drafts of the paper, and approved the final draft.
- Liming Xue performed the experiments, prepared figures and/or tables, and approved the final draft.
- Xiting Huang conceived and designed the experiments, authored or reviewed drafts of the paper, and approved the final draft.

## Human Ethics

The following information was supplied relating to ethical approvals (i.e., approving body and any reference numbers):

This study was approved by the Ethics Committee of Southwest University (IRB NO. H18007), and all involved procedures conformed to the tenets of the Declaration of Helsinki.

## Data Availability

The data is available at Figshare: Yue, Tong (2020): Raw data.zip. figshare. Dataset. https://doi.org/10.6084/m9.figshare.12170445.v1.

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
