# Peer review of "Oxytocin weakens self-other distinction in males during empathic responses to sadness: an event-related potentials study"

_PeerJ, doi:10.7717/peerj.10384_

## Round 0.1 · original submission · Major Revisions

Dear Authors,

Please do the necessary revisions.

Reviewer 1 ·

Basic reporting

The manuscript “Oxytocin weakens self-other distinction in males during empathic responses to sadness: An event related potentials study” focuses on a very interesting research question. However, as detailed below, I have several main concerns regarding this manuscript:
- Introduction. The conceptual framework in the introduction is very limited. The authors should include relevant literature on empathy and self-other distinction at implicit and explicit level as well as on oxytocin and self-other distinction. In addition, the authors would need to justify the rationale for selecting sad facial expressions based on previous studies on oxytocin and empathy. The authors are invited to revise the manuscript to remove all the misleading information (e.g., introduction: “extreme masculine cognitive style”. The authors did not measure it; the references to autism is misleading. No measures of autism traits were included).
- The Methods section needs more details. “We believed that the peptide would increase its concentrations in the cerebrospinal fluid during this time.” Please include relevant references with respect to the time elapsing between drug administration and testing.

- I commend the authors for controlling baseline group differences with respect to IRI,; however, they do not consider possible important group differences with respect to the affective state/mood and to the awareness of one’s own emotions (self-awareness/alexithymia). It is difficult to make strong inferences without these controlling measures.
The manuscript should be revised throughout for clarity. A few examples: abstract: “..influence the normally high level of self-other distinction”, what does “normally high levels” mean?; introduction “problem”, perhaps “issue” or “question”? Methods:” the other–task, we told the subjects to press the F button if the emotion expressed by the face was sad and the F button if neutral” “F” and “F” ? Discussion: “Although the predecessors claimed that OXT could reduce the difference wave in the self-task by decreasing brain reactivity to negative emotion and increasing neural activity involved in emotion regulation (Ma et al., 2016), it's hard to explain” Predecessors?

The authors claim that oxytocin reduces self-other distinction. The discussion section would be strengthened by situating the results of the current study within the context of studies highlighting that oxytocin affects self-other distinction at explicit level but not at implicit level (e.g., Pfundmair, M., Rimpel, A., Duffy, K., & Zwarg, C. (2018). Oxytocin blurs the self-other distinction implicitly but not explicitly. Hormones and behavior, 98, 115-120. Colonnello, V., Chen, F. S., Panksepp, J., & Heinrichs, M. (2013). Oxytocin sharpens self-other perceptual boundary. Psychoneuroendocrinology, 38(12), 2996-3002). In addition, I suggest expanding the limitation section: the sample size is very small and no measures of self-awareness and mood were included.

Experimental design

The research question is not well defined. I suggest the authors include relevant literature on empathy and recognition of sad expressions.
Overall, the methods are well described. However additional references are needed (e.g., drug administration procedure)

Validity of the findings

The conclusions should be revised. The results should be considered in the context of the current literature on oxytocin and self-other distinction.

·

Basic reporting

The article is well-written and places the present study and results in the appropriate context of prior studies.

Experimental design

The experimental design is sound, although the samples sizes could be larger.

Validity of the findings

Despite the small sample sizes, there were several significant findings. This work builds on previous studies on this topic and addresses important questions. The conclusions are strongly supported by the present results without excessive speculation about their relevance and impact.

Additional comments

1. There is a need for more introductory material on the merits of ERP and general background on the technique for those not experts in this area. It should be made clear when discussing fMRI vs. ERP findings. and the relationship between these findings should be explained.

2. The discussion would be strengthend by a brief summary of the findings in the first paragraph, beyond just noting differences which are discussed in later paragraphs.

3. Were there any analyses evaluating interactions between IRI values and the treatment effects? This might provide additional useful findings.

4. It would seem that these findings could be relevant beyond autism. What other populations, conditions, or disorders might these findings be relevant for? It would seem important to connect these findings to maternal behavior topics given the common use of OXT in the peripartum setting. While the present study only includes males, they all had mothers.

---

## Round 0.2 · Minor Revisions

Dear Author,Our peer reviewer has requested minor remaining revisions to the submitted revised manuscript.Thanking you.

Reviewer 1 ·

Basic reporting

The quality of the manuscript has significantly improved. However, I have a few more concerns and comments:
Abstract: “to investigate whether OXT can weaken men’s excessive self-other distinction”. Is it really “excessive”? Based on what? Please clarify or delete here and in the text of the manuscript.
Introduction: “men display a lesser degree of concern about others than women because of their higher intrinsic psychological significance of self relevance information”. Please clarify this sentence and its relation to the self-task, if any.
Conclusions: “indicating OXT’s role in integrating the self with others instead of separating them.” This statement is very strong. Please revise it or elaborate a bit more on it.
It looks like that an additional round of English editing is needed (e.g., “Of these 40, one subject withdrew at the drug application stage because of rhinitis, leaving 39 subjects remaining.” …leaving and remaining?)
Please upload a more recent certificate of English editing.

Experimental design

The experimental design and the methods are described with sufficient information.

Validity of the findings

No additional changes are required

---

## Round 0.3 · accepted · Accept

Dear Authors, I have read your rebuttal and checked your revised manuscript and find that you have revised according to the comments by the peer reviewers. Thank you for your effort.

·

Basic reporting

No comment

Experimental design

No comment

Validity of the findings

No comment

Additional comments

No comment